# Work Measurement in OPEN Quantum System

**DOI:** 10.3390/e24020180

**Published:** 2022-01-25

**Authors:** Youyang Xu

**Affiliations:** Faculty of Science, Kunming University of Science and Technology, Kunming 650500, China; xyynx1981@gmail.com

**Keywords:** open quantum system, measurement of work, Jarzynski equality

## Abstract

Work is an important quantity in thermodynamics. In a closed quanutm system, the two-point energy measurements can be applied to measure the work but cannot be utilized in an open quantum system. With the two-point energy measurements, it has been shown that the work fluctuation satisfies the Jarzynski equality. We propose a scheme to measure the work in an open quantum system through the technique of reservoir engineering. Based on this scheme, we show that the work fluctuation in open quantum system may violate the Jarzynski equality. We apply our scheme to a two-level atom coupled to an engineered reservoir and numerically justify the general results, especially demonstrating that the second law of thermodynamics can be violated.

## 1. Introduction

Quantum thermodynamics is developing very fast and covers the exploration from few-body to many-body quantum physics and non-equilibrium statistical mechanics [1,2]. Being different from the traditional quantum statistics, where the concepts such as boson/fermion statistics play important roles, in quantum thermodynamics, quantum coherence and quantum correlation are involved [3,4,5], which originate from quantum information science [6,7,8]. In particular, quantum fluctuations can be observed in conventional thermodynamic quantities, e.g., work and heat [9,10,11,12,13].

For a driven system, Alicki defines the work as the energy change of the system induced by changing the Hamiltonian W=∫TrρdH, with ρ and *H*, respectively, being the density matrix and the Hamiltonian of the system [14] and further generalized by Boukobza and Tannor to the definitions of heat flux and power [15]. We note that these definitions are only applied to the weak coupling between the system and the heat reservoir [16]. From the point of view of operation, to measure the work, the state of open quantum system should be monitored continuously. However, once the system is measured, its state is changed by the interaction between the system and the measurement pointer, and as a result, the work defined above can not be measured. To overcome this problem, a two-point energy measurement or generalized quantum measurement are invented [12,17], i.e., in an isolated system the work is equal to the change of the internal energy, i.e., $W=\Delta U=Tr\rho^{f}H^{f}-Tr\rho^{i}H^{i}$, with the superscript *f* and *i* representing the final and initial times of the process, respectively. Although this method has been applied to demonstrate many important conclusions in quantum thermodynamics, such as to study the fluctuation relations of the work of quantum driven system [18], it can only be used in the case of a closed system. These two kind definitions are consistent with each other on the level of the ensemble but different on the level of the system. In addition, their different origins from that of the two-point energy measurements will destroy the coherence of the system.

As the work has the characteristic of probability, its fluctuation theorem has attracted much attention. In a classical non-equilibrium system, it has been shown that the Jarzynski equality $\langle e^{-\beta w}\rangle =e^{-\beta \Delta F}$ is satisfied by the work, from which the second law of thermodynamics can be deduced [19]. Then, with the two-point energy measurements, it has been shown that this equality can also be applied to closed quantum system [13], but with generalized measurement, the Jarzynski equality can be violated [20]. With the total system being viewed as a closed system, the Jarzynski equality has been generalized to arbitrary open quantum systems, and the free energy should be changed correspondingly [21]. Alternatively, through a one-time measurement of the initial energy and the measurement of heat, a modified quantum Jarzynski equality of the guessed quantum work can be obtained [22]. We note that there are two problems in the Jarzynski equality of arbitrary open quantum systems. The first is that the work of a open quantum system cannot be measured through the two-point energy measurements or the scheme in [22], as the work is not equal to the energy change of the system. The second is that the free energy of the system is interaction-dependent, i.e., the free energy depends on the interaction between the system and its heat reservoir. As a result, the free energy cannot be measured directly. These two problems result in the Jarzynski equality of open quantum systems being unable to be explored experimentally.

We note that both the above problems originate from the interaction with the heat reservoir. As such, it may be possible to overcome these problems through reservoir engineering. This technique has become an important technology in quantum science. With the reservoir engineering, the exotic phases of quantum matter can be simulated quantumly [23,24,25], thermal averages can be computed in a variety of fields, such as in statistical mechanics [26,27] and machine learning [28], and the quantum non-Markovian behavior can be measured [29]. It has been shown that the effect of the reservoir can also be simulated with classical noise regardless of Markovian or non-Markovian dynamics [30,31], which will be applied to overcome the problems encountered by the work measurement of an open quantum system.

In this paper, we will propose a scheme to measure the work of an open quantum system through reservoir engineering. Then, based on this scheme, we obtain a fluctuation theorem of the work and show that it is different from the Jarzynski equality. At last, the experimental feasibility based on a single trapped ion is provided and shows that the second law of thermodynamics presented by W≥ΔF can be violated.

## 2. Work Measurement of an Open Quantum System

We consider that an open quantum system is driven by forcing and simultaneously interacting with an engineered reservoir. The reservoir engineering has been implemented in trapped ions, for example, the high-temperature amplitude reservoir can be achieved by applying random electric fields along the axis of the trap, and the zero-temperature reservoir can implemented by a cooling laser applied to the ions [32]. Being similar to the high-temperature amplitude reservoir, we assume that the interaction between the system and the reservoir can be described by a classical stochastic process. For simplicity, we assume that there is only one noise, and our following results can easily be generalized to the case containing multi-noise, which can generate the non-Markovian dynamics. With this engineered reservoir, the state evolution of the system can be described by a stochastic quantum Liouville equation [31]:(1)$$\frac{d\rho_{st}\left(t\right)}{dt}=-i\left[H\left(t\right),\rho_{st}\left(t\right)\right]-i\sqrt{\gamma }\left[\eta \left(t\right)A,\rho_{st}\left(t\right)\right],$$
where $\rho_{st}\left(t\right)=|\phi (t)\rangle \langle \phi (t)|$ is the stochastic density matrix corresponding to one realization of the Gaussian processes η(t), H(t) is the driven Hamiltonian by a work source, γ is a positive real constant and depends on the coupling strength between the system and the reservoir, and *A* is a Hermitian operator. Here, for simplicity, the classical stochastic process η(t) is a white noise process satisfying $\langle \eta (t)\rangle =0$ and $\langle \eta \left(t\right)\eta \left(t^{\prime }\right)\rangle =\delta \left(t-t^{\prime }\right)$, with $\langle \eta (t)\rangle$ representing averaging over noise realizations. The noise-averaged density matrix $\rho \left(t\right)=\langle \rho_{st}\left(t\right)\rangle \;$ is obtained by averaging over different realizations $|\phi \rangle \langle \phi |$, and correspondingly, the master equation induced by the corresponding reservoir can be obtained, i.e., $\frac{d\rho (t)}{dt}=-i\left[H\left(t\right),\rho \left(t\right)\right]+D\left[\rho \left(t\right)\right]$, where D[ρ(t)] is the dissipator due to the reservoir [31]. Although the direct measurement of the state will prevent the measurement of the work as mentioned above, it can be obtained as follows. We note that the state evolution of the Equation (Equation 1) can also be described by its corresponding stochastic unitary operator:(2)$$U_{0\to t}\left[\eta \left(t\right)\right]=T\exp\left[-i\int_{0}^{\tau }H_{st}\left(t^{\prime }\right)dt^{\prime }\right]$$
where T represents a time-ordering operator, and $H_{st}\left(t^{\prime }\right)=H\left(t^{\prime }\right)+H_{st}^{I}\left(t^{\prime }\right)$ is the stochastic Hamiltonian, with $H_{st}^{I}\left(t^{\prime }\right)=\sqrt{\gamma }\eta \left(t^{\prime }\right)A$ being the interaction of the Hamiltonian with the noise. In such an open system, the stochastic unitary operator $U_{0\to t}\left[\eta \left(t\right)\right]$ can be viewed as a functional of the noise η(t), see Appendix A. This result implies that the time evolved state of the system can be obtained by continuously monitoring the noise, i.e., the state of the open system at time *t* is $|\phi (t)\rangle =U_{0\to t}\left[\eta \left(t\right)\right]|\phi (0)\rangle$.

Based on the obtained state, the quantum work in a quantum trajectory can be calculated as:(3)$$w_{k}=\int \langle \phi_{k}\left(t\right)|dH|\phi_{k}\left(t\right)\rangle ,$$
where the Hamiltonian is determined by the driving protocol, the state is $|\phi_{k}\left(t\right)\rangle =U_{0\to t}\left[\eta \left(t\right)\right]|\phi_{k}\left(0\right)\rangle$, and the initial state $|\phi_{k}\left(0\right)\rangle$ is chosen from the initial ensemble $\rho \left(0\right)=\sum p_{k}|\phi_{k}\left(0\right)\rangle \langle \phi_{k}\left(0\right)|$, with $p_{k}$ being the corresponding probability of the state $|\phi_{k}\left(0\right)\rangle$. The averaged work fluctuation $\langle \langle e^{-\beta w[\eta (t)]}\rangle \rangle$ can be obtained through the twice average, i.e., the average $\langle e^{-\beta w_{k}\left[\eta \left(t\right)\right]}\rangle =\int p\left[\eta \left(t\right)\right]e^{-\beta w_{k}\left[\eta \left(t\right)\right]}d\eta \left(t\right)$ with respect to the noise η(t) with p[η(t)] being probability density of the noise η(t), and the initial ensemble average $\langle e^{-\beta w[\eta (t)]}\rangle =\sum p_{k}e^{-\beta w_{k}\left[\eta \left(t\right)\right]}$ with respect to the initial state ρ(0). When the noise is small, the functional $U_{0\to t}\left[\eta \left(t\right)\right]$ can be obtained through functional Taylor series $U_{0\to t}\left[\eta \left(t\right)\right]=U_{0\to t}\left[0\right]+\sum \frac{1}{n!}\int \ldots \int \frac{\delta^{n}U_{0\to t}\left[\eta \left(t\right)\right]}{\delta \eta \left(t_{1}\right)\cdots \delta \eta \left(t_{n}\right)}\eta \left(t_{1}\right)\cdots \eta \left(t_{n}\right)dt_{1}\cdots dt_{n}$, where the zeroth order is $U_{0\to t}\left[0\right]=$$T\exp[-i\int_{0}^{t}H\left(t^{\prime }\right)dt^{\prime }]$, and the term $\frac{\delta^{n}U_{0\to t}\left[\eta \left(t\right)\right]}{\delta \eta \left(t_{1}\right)\cdots \delta \eta \left(t_{n}\right)}$ is the *n*th functional derivative [33]. In our following numerical simulation, we cut off the functional Taylor series up to the second order, and their expressions are provided in Appendix A. Using this functional Taylor series, we can also expand the work $w_{k}\left[\eta \left(t\right)\right]=w_{k}\left[0\right]+\delta w_{k}\left[\eta \left(t\right)\right]+\delta^{2}w_{k}\left[\eta \left(t\right)\right]$ as a functional Taylor series with $\delta^{n}w_{k}\left[\eta \left(t\right)\right]$ representing the nth order variation, and its form is shown in Appendix A. The work at the zeroth order $w_{k}\left[0\right]$ is equal to the work of the corresponding closed system (corresponding to the case of zero noise η(t)≡0), i.e.,
(4)$$w_{k}\left[0\right]=E_{k}^{\tau }-E_{k}^{0},$$
where the internal energy is $E_{k}^{t}=\langle E_{k}\left(t\right)|H\left(t\right)|E_{k}\left(t\right)\rangle$ with the state $|E_{k}^{t}\rangle =U_{0\to t}\left[0\right]|\phi_{k}\left(t\right)\rangle$, and this is the first law of thermodynamics in closed system. This result shows that the coherence between the eigen-states of the Hamiltonian induce the difference between the work defined Equation (Equation 3) and that through the two-point energy measurements. With the properties of the white noise $\langle \eta (t)\rangle =0$ and $\langle \eta \left(t\right)\eta \left(t^{\prime }\right)\rangle =\delta \left(t-t^{\prime }\right)$ and up to the second order, we find that only the following two fluctuations have contributions to the fluctuation theorem, i.e., $\langle \delta^{2}w_{k}\left[\eta \left(t\right)\right]\rangle$ and $\frac{1}{2}\langle \delta w_{k}\left[\eta \left(t\right)\right]\delta w_{k}\left[\eta \left(t\right)\right]\rangle$, please see Appendix A.

## 3. Work Fluctuation Theorem of Open Quantum System

Firstly, we demonstrate a work fluctuation theorem in the corresponding closed system. Here and following, the initial state is chosen to be the Gibbs state $\rho \left(0\right)=\frac{1}{Z_{0}}e^{-\beta H(0)}$, with β=1/T being the inverse temperature, the Boltzmann’s constant being set as $k_{B}=1$, and $Z_{0}=Tr\left[e^{-\beta H(0)}\right]$ being the partition function. In a closed system and through two-point energy measurements, it has been shown that the work fluctuation satisfy the following theorem proposed by Jarzynski, $\langle e^{-\beta w[0]}\rangle =e^{-\Delta F}$, with F=−TlnZ being the free energy and $Z=Tre^{-\beta H(t)}$ being the partition function. Being different from this result, it can be shown that, through the definition in Equation (Equation 3), the fluctuation theorem of the work in a closed system is:(5)$$\langle e^{-\beta w[0]}\rangle =e^{-\beta \Delta f},$$
where the quantity $f\left(t\right)=-T\ln\sum_{k}e^{-\beta E_{k}\left(t\right)}$ (here and following we call it quasi-free energy, which has been defined as free energy in [22]) can be viewed as free energy when the state $|\phi_{k}\left(t\right)\rangle$ is the eigen-state of the Hamiltonian H(t). In other words, Equation (Equation 5) is equivalent to the Jarzyski equality in a quasistatic process, but generally, they are different. There is a special case where the state $|\phi_{k}\left(t\right)\rangle$ can always be the eigen-state of the Hamiltonian H(t), if the Hamiltonian is a conserved quantity or, equivalently, the following commutator is satisfied: $[H\left(t\right),H\left(t^{\prime }\right)]=0$. Correspondingly, the quasi-free energy is equal to the free energy in this case. The Equation (Equation 5) can be demonstrated as follows. Using the first law of thermodynamics in a closed system, i.e., with Equation (Equation 4), the fluctuation of work in this closed system can be reduced to $\langle e^{-\beta w[0]}\rangle =\sum_{k}p_{k}e^{-\beta [E_{k}\left(\tau \right)-E_{k}\left(0\right)]}=\sum_{k}e^{-\beta E_{k}\left(\tau \right)}/\sum_{k}e^{-\beta E_{k}\left(0\right)}=e^{-\beta \Delta f}$, with $p_{k}=e^{-\beta E_{k}\left(0\right)}/\sum_{k}e^{-\beta E_{k}\left(0\right)}$ being the probability of the initial energy $E_{k}\left(0\right)$, which completes the demonstration. When the noise is small, this fluctuation theorem can be viewed as the zeroth approximation of the open system.

Now we show how to obtain the work fluctuation theorem in an open quantum system. Rewriting the work as $w_{k}\left[\eta \left(t\right)\right]=E_{k}\left(\tau \right)-E_{k}\left(0\right)+\Delta w_{k}\left[\eta \left(t\right)\right]$, with Δw[η(t)] being the finite correction of work due to the increase in the heat reservoir, we can formally write the fluctuation theorem of the open system as the Jarzyski-like equality:(6)$$\langle \langle e^{-\beta w[\eta (t)]}\rangle \rangle =e^{-\beta \Delta f^{\prime }},$$
where $f^{\prime }\left(t\right)=-T\sum_{k}e^{-\beta [E_{k}\left(t\right)+\Delta E_{k}\left(t\right)]}$, with the energy correction $\Delta E_{k}=-T\ln\langle e^{-\beta \Delta w_{k}\left[\eta \left(t\right)\right]}\rangle$ being noise averaged. This result means that the second law of the thermodynamics represented by the Jarzyski equality can be invalid. We note that the quantity $f^{\prime }\left(t\right)$ is different from the quasi-free energy, as it is noise-dependent. Now we consider the case that the noise is weak. In such case, we can approximate $e^{-\beta w_{k}\left[\eta \left(t\right)\right]}=e^{-\beta w_{k}\left[0\right]}\left[1-\beta \delta w_{k}\left[\eta \left(t\right)\right]-\beta \delta^{2}w_{k}\left[\eta \left(t\right)\right]+\frac{1}{2}{\left(\beta \delta w_{k}\left[\eta \left(t\right)\right]\right)}^{2}\right]$, and then Equation (Equation 6) can also be approximated as:(7)$$\langle \langle e^{-\beta w[\eta (t)]}\rangle \rangle =e^{-\beta \Delta f}\left(1-\Delta \zeta \right).$$
where the correction is $\Delta \zeta =\sum_{k}p_{k}^{\prime }\left(\beta \langle \delta^{2}w_{k}\left[\eta \left(t\right)\right]\rangle -\frac{1}{2}\beta^{2}\langle \delta w_{k}\left[\eta \left(t\right)\right]\delta w_{k}\left[\eta \left(t\right)\right]\rangle \right)$, with $p_{k}^{\prime }=e^{-\beta E_{k}\left(\tau \right)}/\sum_{k}e^{-\beta E_{k}\left(\tau \right)}$ being the probability of the energy $E_{k}\left(\tau \right)$ at final time τ, as shown in Appendix A. For white noise, this correction is independent on the noise η(t). This result shows more obviously that the second law of thermodynamics can be violated in the case of that the Hamiltonian is a conserved quantity, and the correction Δζ is larger than zero. In the following, we will demonstrate this result numerically. For our following comparison, we denote $\zeta_{1}=\langle \langle e^{-\beta w[\eta (t)]}\rangle \rangle$, $\zeta_{2}=e^{-\beta \Delta f}\left(1-\Delta \zeta \right)$ and $\zeta_{3}=$$e^{-\Delta F}$, correspondingly.

To solve the violation of the second law, we note from the Equation (Equation 7) that the free energy should be redefined. For this purpose, we add the noise to our system to compose an adiabatic system. In this composite system, the total energy is $E_{k}^{tot}\left(\tau \right)=E_{k}\left(\tau \right)+E_{r}\left(\tau \right)$, where we introduce the phenomenologically $E_{r}\left(\tau \right)$ as the noise energy, from which the work can be obtained:(8)$$w_{k}\left[\eta \left(t\right)\right]=E_{k}^{tot}\left(\tau \right)-E_{k}^{tot}\left(0\right),$$
which is the law of energy conservation. Based on this result, it can be shown that the following Jarzynski equality can be obtained:(9)$$\langle \langle e^{-\beta w[\eta (t)]}\rangle \rangle =e^{-\beta \Delta f^{p}},$$
where we redefine the free energy as $f^{p}\left(\tau \right)=\langle \sum_{k}e^{-\beta [E_{k}\left(\tau \right)+E_{r}\left(\tau \right)]}\rangle$. As a result, the Jarzynski equality (or the second law expressed with this equality) is reserved by this redefined free energy. We note that this definition of the free energy is very similar as that defined for the open quantum system at strong coupling with reservoir, where the reservoir is viewed as a quantum system [21]. We also note that Equation (Equation 9) is a theoretical result and cannot be demonstrated experimentally now, as the phenomenological noise energy $E_{r}\left(\tau \right)$ cannot be measured with the current technology. This problem also arises in the quantum thermodynamics at strong coupling [21].

## 4. Experimental Feasibility

Based on the above results, in the following we propose how to observe the fluctuation theorem of work in a single ultracold ${}^{40}Ca^{+}$ ion stably confined in a Paul trap [34], and our proposal can also be applied to other systems. In this system, the engineered reservoirs have been explored experimentally [35]. As in [36], the system is a Zeeman qubit encoded in two electronic levels of the ${}^{40}Ca^{+}$ ion, i.e., $|↓\rangle =|4^{2}S_{1/2},m_{J}=-1/2\rangle$ and $|↑\rangle =|4^{2}S_{1/2},m_{J}=+1/2\rangle$ (with the transition frequency ω(t)). Usually, this transition frequency ω(t) is determined by a static magnetic field, but here we can introduce a time-dependent magnetic field through which the transition frequency $\omega \left(t\right)=\omega_{0}+kt$ can be controlled to vary with time. The Hamiltonian of the driven qubit is:(10)$$H\left(t\right)=\frac{1}{2}\omega \left(t\right)\sigma_{z}.$$In this protocol, we have $[H\left(t^{\prime }\right),H\left(t\right)]=0$, which, as mentioned above, will result in the quasi-free energy being equal to the free energy (or equivalently, $\zeta_{3}=e^{-\beta \Delta f}$), and the violation of the second law can be demonstrated directly through Equation (Equation 7). The heat reservoir is a fluctuating field $B=B_{0}\eta \left(t\right)$ (which can be generated by an RF noise generator being similar to that demonstrated in Ref [32]), with $B_{0}$ here being the amplitude of the fluctuating field. The interaction between the system and the fluctuating field can be described as:(11)$$H_{st}^{I}\left(t\right)=\sqrt{\gamma }\eta \left(t\right)\sigma_{x},$$
where γ is determined by the amplitude of the fluctuating field and can be obtained through Rabi oscillation in the experiment.

Our proposal is very similar to those used to demonstrate the Jarzynski equality in closed systems, i.e., including four steps. The first step is preparing the state of the ion in a thermal state $\exp\left[-\beta H\left(0\right)\right]/Z_{0}$ with the inverse temperature being defined as $\beta =-\frac{1}{\omega_{0}}\ln\frac{p_{↑}^{i}}{p_{↓}^{i}}$, with $p_{↓}^{i}$ and $p_{↑}^{i}$ being the initial populations of the state $|↓\rangle$ and $|↑\rangle$, respectively. This initial thermal state can be obtained by preparing the qubit to the state $|↓\rangle$, then to a superposition state $\sqrt{p_{↓}^{i}}|↓\rangle +\sqrt{p_{↑}^{i}}|↑\rangle$ through the carrier-transition and waiting a desired time (the dephasing time of the qubit). In the second step, the ion is measured to project to an energy eigen-state $|↓\rangle$ or $|↑\rangle$. Through this step, the initial free energy can be obtained through the relation $p_{↑}^{i}=\frac{1}{Z_{0}}e^{-\frac{1}{2}\beta \omega_{0}}$ and $p_{↓}^{i}=\frac{1}{Z_{0}}e^{\frac{1}{2}\beta \omega_{0}}$. In the third step, the driving force and the fluctuating field are applied, and simultaneously, the fluctuating field is monitored continuously. Then, the work can be obtained by the driving protocol and the monitored fluctuating field, and the correction Δζ included in the term $\zeta_{2}$ can be calculated by the driving protocol as shown in Appendix A. At the last step, another projective measurement of the Hamiltonian H(T) is made to obtain the final free energy, which can be achieved by another projective measurement to the energy eigen-state $|↓\rangle$ and $|↑\rangle$.

In the following, we simulate numerically this driving protocol. The stochastic Hamiltonian is $H_{st}\left(t\right)=\frac{1}{2}\omega \left(t\right)\sigma_{z}+\sqrt{\gamma }\eta \left(t\right)\sigma_{x}$. In Figure 1a, we show the case of a weak fluctuating field. Obviously, when the temperature is high, the right hand side of the Equation (Equation 7) is well coincided with the left side, and the violation of the second law can be observed, which is implied by the $\zeta_{1}$ being larger than $\zeta_{3}$, but this violation will disappear when the temperature is increased. In Figure 1b, we show the relations between $\zeta^{\prime }$s and the coupling strength at medium temperature. It shows that the approximation in Equation (Equation 7) is satisfactory when the coupling between the system and its heat reservoir is weak and that the violation of the second law is more obvious when the coupling become stronger. These results imply that the usual definition of the free energy is unsuitable for the description of quantum thermodynamics process.

## 5. Conclusions

Work is an important quantity in thermodynamics, but it could not be measured directly in a open quantum system before. In this paper, we provided a scheme to overcome this problem. In a closed quantum system, we showed that the free energy should be replaced by the quasi-free energy, and with this quasi-free energy, the Jarzynski equality is satisfied. However, in an open quantum system, we showed that the Jarzynski equality can be violated. At last, an experimental scheme was provided, and it was demonstrated numerically that the second law of thermodynamics can be violated.

## Figures and Tables

**Figure 1 entropy-24-00180-f001:**
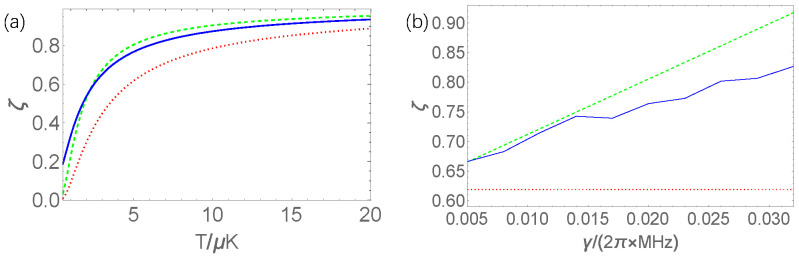
(**a**) The quantities $\zeta_{1},\zeta_{2}$, and $\zeta_{3}$ vs. the temperature *T*, where $\zeta_{1},\zeta_{2}$, and $\zeta_{3}$ are represented by the blue solid, green dashed, and red dotted curves, respectively. In our simulation, the parameters are chosen as $\omega_{0}=20\times 2\pi$ MHz, k=0.2×2π MHz/μs, γ=0.02×2π MHz, and τ=5μs. (**b**) The corresponding quantities in (**a**) vs. the coupling strength γ. The temperature is chosen to be T=5μK, and the other parameters are the same as those in (**a**).

## Data Availability

Not applicable.

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
