# Peer review of "Work Measurement in OPEN Quantum System"

_entropy, 2022, doi:10.3390/e24020180_

Round 1

Reviewer 1 Report

The manuscript deals with the important problem how to measure work in open quantum systems. The scheme, proposed by the author, to achieve a measurement of work through the technique of reservoir engineering is certainly a valuable proposal. However, the statement, that the second law of thermodynamics can be violated, needs further clarification. How should this be possible? Which implications does this have? The second law of thermodynamics is such a fundamental element of theoretical physics that this point needs a much more careful discussion in the manuscript.

Author Response

We thank the reviewer for the constructive criticisms, which have helped us improve our manuscript. All the changes are emphasized with underline. Point-by-point responses to the comments are given below.

However, the statement, that the second law of thermodynamics can be violated, needs further clarification. How should this be possible? Which implications does this have? The second law of thermodynamics is such a fundamental element of theoretical physics that this point needs a much more careful discussion in the manuscript.

Our reply

We want to state that the violation of the second law is due to the definition of the free energy, or in other word, we should redefine the free energy to reserve the second law. This detail clarification is added in our new version, please see the last paragraph in part III.

Reviewer 2 Report

Although the manuscript proposal is very good, there are several points that were not properly discussed. One comes from the treatment of a general problem, in the end only non-Markovian processes are applicable, that is, to correlated deltas. References are not organized numerically in the manuscript. The algebraic part is well grounded, especially with the addition of the appendix.
In the text, the authors generically treat thermodynamic quantities such as work and entopy, as being dynamic variables of the physical system, since in open systems these quantities also vary due to the fact that the statistical matrix is changing over time.
It also presents a certain imprecision of language, as in the last sentence of the conclusion "At last an experimental scheme was provided, and it was demonstrated numerically that the second law of thermodynamics may be violated". Here, despite having a conclusion it is not affirmative, there is a "may be violated".

Author Response

We thank the reviewer for the constructive criticisms, which have helped us improve our manuscript. All the changes are emphasized with underline. Point-by-point responses to the comments are given below.

1. Although the manuscript proposal is very good, there are several points that were not properly discussed. One comes from the treatment of a general problem, in the end only non-Markovian processes are applicable, that is, to correlated deltas. References are not organized numerically in the manuscript.

Our reply

The reason of only dealing with the Markovian process is that this will simplify our calculation and results. Our results show that only the Markovian effect of reservoir can induce the difference of Jarzynski equality between open and closed systems, and furtherly the non-Markovian process can be obtained by replacing the white noise by colored noise. Additionally, we have corrected the number of the references.

2. It also presents a certain imprecision of language, as in the last sentence of the conclusion "At last an experimental scheme was provided, and it was demonstrated numerically that the second law of thermodynamics may be violated". Here, despite having a conclusion it is not affirmative, there is a "may be violated".

Our reply

We have checked our language again, and especially replaced the word “may” by “can” in your mentioned sentence.

Round 2

Reviewer 2 Report

The authors gave the manuscript a review and improvement as expected. In the current form, the manuscript is more consistent and has a more suitable form for publication.